# Backmapping triangulated surfaces to coarse-grained membrane models

Weria Pezeshkian[1✉], Melanie König[1], Tsjerk A. Wassenaar[1] & Siewert J. Marrink[1✉]

Many biological processes involve large-scale changes in membrane shape. Computer simulations of these processes are challenging since they occur across a wide range of spatiotemporal scales that cannot be investigated in full by any single current simulation technique. A potential solution is to combine different levels of resolution through a multiscale scheme. Here, we present a multiscale algorithm that backmaps a continuum membrane model represented as a dynamically triangulated surface (DTS) to its corresponding molecular model based on the coarse-grained (CG) Martini force field. Thus, we can use DTS simulations to equilibrate slow large-scale membrane conformational changes and then explore the local properties at CG resolution. We demonstrate the power of our method by backmapping a vesicular bud induced by binding of Shiga toxin and by transforming the membranes of an entire mitochondrion to near-atomic resolution. Our approach opens the way to whole cell simulations at molecular detail.

[1] Groningen Biomolecular Sciences and Biotechnology Institute and Zernike Institute for Advanced Materials, University of Groningen, Groningen, Netherlands.
✉email: w.pezeshkian@rug.nl; s.j.marrink@rug.nl

I t may be true that the primary function of biological membranes is their selective permeability that separates the internal cell environment from the outer region. However, it is now clear that many cellular processes take place at the surface of the membranes, and they contribute not only as a substrate but also as an important element of the machinery that is vital for cell survival[1–3]. One of the essential features of these fascinating supramolecular aggregates is their ability to undergo large-scale morphological shape changes. In fact, the shape of biological membranes is continuously adapted to accommodate critical cellular processes, e.g., cell division, vesicular transport, or endocytosis and exocytosis[4–8].

Membrane remodeling processes involve multiple phenomena that each take place at a different time and length scale, ranging from local protein binding and clustering to global membrane shape deformation[3,9]. Therefore, computer modeling of these processes has remained challenging since each method is optimized at a specific scale[10,11]. All-atom molecular dynamics simulation (AA-MD) are able to capture detailed molecular interactions but are typically only feasible for short times and small length scales (around 100–1000 ns and 10–50 nm)[11]. To model phenomena that span a larger scale, up to three orders of magnitude higher than AA-MD, coarse-grained (CG) models are used at the expense of molecular details[12–14]. Nevertheless, AA-MD and CG-MD together do not cover the full scale that is required to investigate membrane-remodeling processes. This shortcoming of AA-MD and CG-MD has pushed the modeling community to use continuum models that are based on macroscopic phenomenological equations that involve only a few model parameters, e.g., dynamical triangulated surfaces (DTS) simulations[15–19]. DTS and similar approaches are indeed successful in describing the shape of simple membranes[17,20–22], but their predictions are limited for realistic cellular membranes as the precise molecular ingredients (protein–protein and protein–lipid interactions) become essential. To overcome these discrepancies, continuum techniques have been coupled in a multiscale scheme, in which the predictions of high-resolution models (i.e., AA or CG-MD) provide inputs for the large-scale model[23–26].

A problem of such bottom-up multiscale simulation approaches is that the input parameters of DTS do not change during the evolution of the system. This implies that the large changes of the membrane shape do not affect the local properties of the membrane. However, unambiguously in complex membranes, the local properties of chemical constituents are strongly affected by large-scale membrane deformations[27]. As an attempt to resolve this limitation, we have developed an algorithm, coined TS2CG (Supplementary Note), that backmaps a DTS structure to a CG representation. The TS2CG algorithm allows us to use DTS to equilibrate the slow and large-scale system evolution and then explore local properties and dynamics at CG resolution. A similar idea has been used to reconstruct a single-site-per-lipid CG representation from a mesoscopic continuum model of membranes remodeled by N-BAR proteins[28]. For the CG part, here we focus on the more detailed Martini model[29], widely applied in the field of membrane simulations[10]. However, our TS2CG algorithm is generically applicable to any CG lipid force field[30–32]. In addition, TS2CG can also be used as a CG membrane builder and solves the limitation of previous methods by generating a smooth user-defined complex membrane with arbitrary shapes[33–35].

In the remainder of this article, we first describe the backmapping algorithm and then show its capacity to model two large-scale membrane remodeling processes, namely the initial step of vesicle fission and membrane bud formation upon Shiga toxin (STxB) binding. We further demonstrate the power of our method by backmapping the highly curved membranes of an entire mitochondrion, resulting in a system composed of more

than 5 million lipids comprising 80 million particles. In the end, we discuss possible routes for further development.

## Results

**Backmapping algorithm**. In this section, we describe the TS2CG algorithm that we developed to efficiently backmap a configuration obtained from a macroscopic DTS simulation to a CG configuration based on the Martini model[29,36]. A DTS structure is defined by $N_v$ vertices, $N_T$ triangles, $N_L$ links which together form an irregular planar triangulated network. Each vertex corresponds to a patch of membrane containing 100 s of lipids and at most one membrane protein[15,24]. In our algorithm, instead of placing a flat segment of a lipid bilayer on each triangle (as it is done in the LipidWrapper software)[35], we increase the number of TS vertices so that the new TS (extended TS) inherits the curviness of the surface. This will enhance the equilibration for two reasons. First, it will avoid the steric contact of the lipids from two neighboring triangles by generating a smooth gradient of normal vectors. Second, the membrane shape inherits the geometrical properties of the DTS simulation, which dominates the bending energy of the membrane, and therefore, no further large-scale shape equilibration is required.

TS2CG makes use of geometrical properties of the TS at each vertex (normal vector and principal curvatures) and therefore we need to evaluate these properties. There are different algorithms to obtain geometrical properties of a closed triangulated surface[17,19,37,38], however, we have used an approach described in ref. [16]. This scheme allows us to obtain a Darboux frame on each vertex, which axes are the normal vector $\hat{\mathbf{N}}(v)$ and two principal directions ($\hat{\mathbf{e}}_1(v), \hat{\mathbf{e}}_2(v)$), located on the vertex tangent plane ($P_v$), corresponding to the direction of maximum and minimum curvatures ($C_1(v), C_2(v)$). $\mathbf{TGL} = [\hat{\mathbf{e}}_1(v), \hat{\mathbf{e}}_2(v), \hat{\mathbf{N}}(v)]$ makes up a rotation matrix that transforms any point from the global coordinate to the local coordinate on the vertex and, since it is unitary, its transpose performs the inverse transformation ($\mathbf{TLG}^{-1} = \mathbf{TLG}^{T}$). Using these geometrical properties, we can backmap a DTS structure to its corresponding CG model by following four steps, as illustrated in Fig. 1. Note that, for simplicity, we assume that the system only contains one type of lipid and protein. However, similar steps can be designed for complex systems containing more protein and lipid types.

*Step 1: Rescaling and monolayer generation*: In this step, the DTS structure will be resized to the preferred size and an upper and lower monolayer TS will be generated. Resizing will be achieved by multiplying the position of each vertex ($\mathbf{X}_v$) to $\mathcal{R}$ (rescaling parameter) that changes the area of the TS surface $A_{TS} \rightarrow \mathcal{R}^2 A_{TS}$. *Note*: In the multi-scale simulations, $\mathcal{R}$ is determined from the DTS calibration (for more details, see refs. [23,24]). Monolayer TSs will be generated by moving the position of each vertex along the normal vector by half of the membrane thickness ($h$) (Fig. 1).

$$\begin{cases} \mathbf{X}_v^+ = \mathbf{X}_v + h\hat{\mathbf{N}}(v) \\ \mathbf{X}_v^- = \mathbf{X}_v - h\hat{\mathbf{N}}(v), \end{cases} \quad (1)$$

$+(-)$ refers to the upper (lower) monolayer.

*Step 2: Pointillism: extending the upper and lower TS*: In this step, the number of the vertices of each TS (upper and lower surfaces) will be increased to a value larger than the required number of the lipids to fully cover their surfaces ($A^{\pm}/a_l$; $a_l$ is area per lipid (APL)). To do this, we first need to make an approximation of the geodesic connecting two neighboring vertices (geodesic on a hypothetical surface that the TS is representing). Consider two vertices ($v_{-1}, v_1$) that are connected by a vector link $\vec{\mathbf{l}}$ of a size of $l$. The projection of $\vec{\mathbf{l}}$ on each vertex

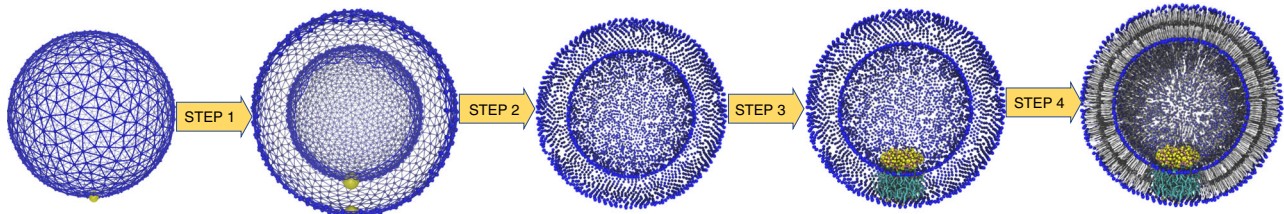

**Fig. 1 Overview of TS2CG backmapping scheme.** (Step 1) A DTS structure of a vesicle containing one protein (yellow bead) is rescaled and two TS structures corresponding to the two monolayers that are generated. (Step 2) Using a Pointillism operation, the number of vertices is increased. (Step 3) The CG protein structure together with a membrane segment is placed at the appropriate TS position. (Step 4) Lipids are placed at the remaining positions and the configuration is ready for subsequent MD simulation.

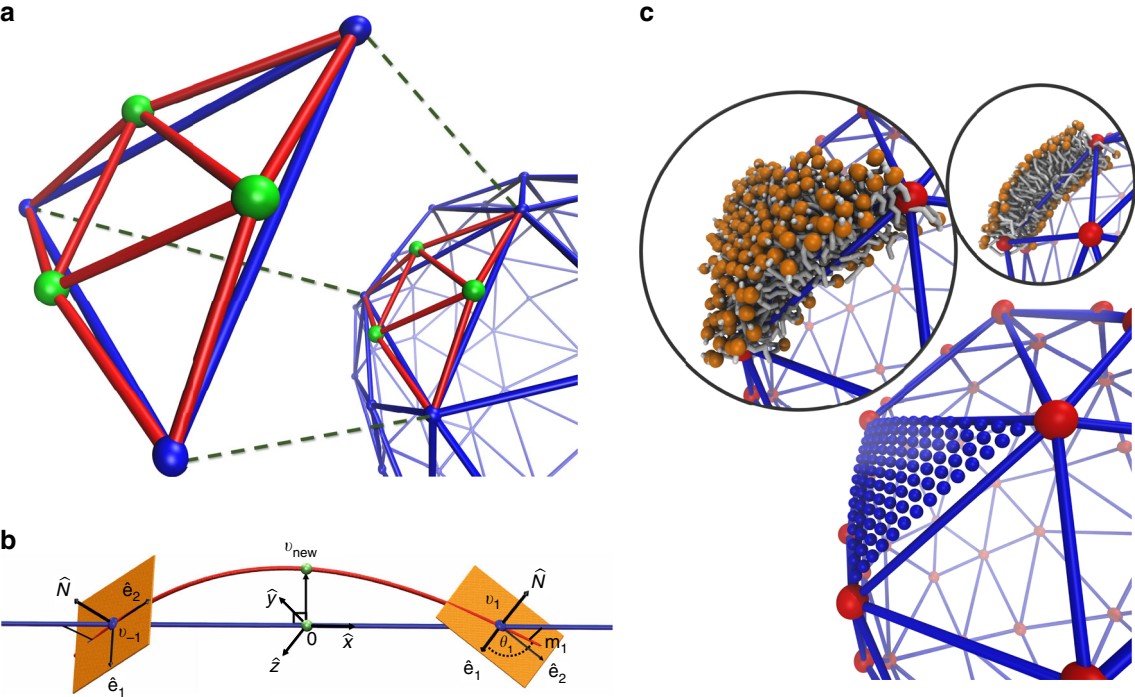

**Fig. 2 Three-dimensional triangulation of a triangle. a** One triangle is divided to four triangles by finding three new vertices in proximity of the mid-point of each of the edges of the original triangle. This increases the vertex number by a factor of 4. **b** Using the geometrical properties at two neighboring vertexes ($v_{-1}, v_1$), an estimate of the geodesic connecting two neighboring vertices can be found (red curve). **c** By performing several consecutive operations as shown in **a**, enough points are generated on each triangle for lipid placement.

plane ($P_{-1}, P_1$) are denoted as $\mathbf{m}_{-1}$, $\mathbf{m}_1$ (Fig. 2b). We require that a geodesic connecting these two vertices ($\vec{\mathbf{s}}_G(t) = 0.5l(t, Y(t), Z(t)); -1 \le t \le 1$) to satisfy three conditions on each vertex (six conditions in total). First, it should cross both vertices, second, it must be parallel to $\mathbf{m}_{-1}$, $\mathbf{m}_1$ when it crosses $v_{-1}, v_1$, respectively, and third, the curvature of the geodesic at $v_i$ must be equal to the TS curvature of $v_i$ in the direction of $\mathbf{m}_i$ (for more details, see Supplementary Methods). Next, for each link in a triangle, we generate a new vertex at $\vec{\mathbf{s}}_G(0)$ and by linking these new vertices, one triangle will be divided to four new triangles (Fig. 2). By performing this operation on all triangles in the TS (Pointillism), the number of vertices is increased by a factor of four. Note: $m$ consecutive Pointillism operations increases the number of vertices as $N_v^{new} \rightarrow 4^m N_v$, and $N_v^{new}$ should be larger than $A^{\pm}/a_l$, therefore, $m > \log_4^{A^{\pm}} - \log_4^{a_l N_v}$. $A$ is a TS total area and $\pm$ refer to the upper/lower layer (Fig. 1).

*Step 3: CG protein placement*: In this step, each CG protein structure is transferred to its given vertices using **TLG**. This will guarantee a correct orientation of the proteins both along the normal and on the local plane. To avoid re-equilibration of lipid–protein interactions, instead of transferring only proteins, one can transfer a protein with its neighboring lipids (including lipids underneath for peripheral membrane proteins) from an equilibrated MD structure (Fig. 1).

*Step 4: CG lipid placement*: We first calculate the number of the required lipids to cover the surface of each monolayer. To do this, we first calculate the total area of the monolayer TS ($A_i; i = 1, 2. i$ is monolayer index) as

$$N_{lipid}^i = \frac{A_i - N_p A_p}{a_l} \tag{2}$$

where $N_p$ and $A_p$ are the number of proteins and projected area of the transmembrane region of the protein (protein projected area for peripheral membrane proteins). Then, we randomly select a vertex, check if it is in touch with any protein, if not, then place a lipid with a probability of $P_{acc} = a_v/a_l$. This step will be iterated until $N_{lipid}^i$ lipids are selected. Our results show a quick convergence of this algorithm (Fig. 1).

**Backmapping vesicular buds**. To check whether our TS2CG backmapping algorithm generates stable CG structures, we test our method on two processes, namely vesicle growth and membrane bud formation upon STxB binding. Further, we use TS2CG to generate mitochondrial membranes (with a realistic size and shape) from electron microscopy tomography (EM) data. In all cases we use the Martini force field as the CG model (Martini 2.2 and Dry Martini force field)[39,40].

The first example deals with vesicle growth, resulting from lipid absorption in the outer monolayer of a vesicle. To mimic vesicle growth using DTS, we increase the mismatch between the inner and the outer monolayer areas ($\Delta A_0$). This is achieved by applying an energy potential, in addition to the standard bending energy, as $E_s = \frac{k_r}{8h^2 A_f}(\Delta A - \Delta A_0)^2$ where $2h$ is the membrane thickness and $k_r$ is the area compression modulus[18]. In addition, we assume that the growth happens fast, and therefore, the inner volume of the vesicle remains constant. This is done by inserting a volume controlling potential as $E_V = \frac{K_v}{2V_f}(V - V_0)^2$ which relaxes the system volume to a fixed final value of $V_0$. In our simulation, we have chosen the membrane bending rigidity $\kappa = 20k_BT$, $K_v = 60k_BT$, $k_r = 3\kappa = 60k_BT$, $\Delta A_0 = 0.3A_0$, $V_0 = 0.7\Upsilon(A_0)$, $A_0$ is the TS total area at the end of the growth and $\Upsilon(A_0)$ is volume of a spherical vesicle with an area of $A_0$ (see Supplementary Methods for more details and the relevance of the input values). With this setup, we performed a DTS simulation and found that the vesicle changes its conformation from a spherical to a dumbbell shape structure (Fig. 3a, b).

Next, we aim to backmap the final DTS structure (Fig. 3b), to its corresponding Martini model of a pure 1,2-dioleoyl-sn-glycero-3-phosphocholine (DOPC) lipid system. To do this, we first need to find a correct rescaling factor ($\mathcal{R}$). In a DTS simulation, the minimum distance between any pair of vertices is equal or larger than $d$ (DTS length metric, see the Methods section and ref. [15] for more detail). Below nm length scale, two

bilayers can repel each other through hydration and membrane protrusion forces that are below the resolution of the DTS but are captured by the Martini model. To account for these forces in DTS, we treat them as an infinite repulsion by forbidding different segments of a membrane surface (like opposing vertices in the neck region of the Fig. 3b) to come closer to a distance in which these forces are effective ($\sim 0.7$ nm)[41,42]. Thus, $R$ is the bilayer thickness ($\sim 3.8$ nm) plus a distance in which hydration and membrane protrusion forces are effective ($\sim 0.7$ nm), meaning $\mathcal{R} = 4.5$ nm/$d(d = 1$ in our DTS simulations). Note that, for membranes with small deformations, $\mathcal{R} \gg 4.5$ nm can also be chosen. However, for highly deformed membranes, this choice implies that many probable configurations (for example the neck region in Fig. 3) have not been visited in the DTS simulation. By setting $\mathcal{R} = 4.5$ nm, we generated two TS surfaces for both monolayers (Step 1). The obtained TS's required three consecutive Pointillism operations ($m = 3$) to generate enough points for lipid placement (Step 2). For this system, Step 3 is not required as the vesicle does not contain any protein. Performing Step 4 resulted in 50,669 (41,500) DOPC lipids in the upper (lower) monolayer. After energy minimization, using the solvent-free Dry Martini force field[39], and 10 ns of equilibration (using position restraints on the phosphate beads to avoid large size pore formation), a 200 ns MD simulation, was performed (see the Methods section for more detail on the MD simulations and for evolution of the system's energy see Supplementary Fig. 1A). The final structure of the MD simulation is shown in Fig. 3c. Note: within the first 50 ns, all the small pores were closed.

**Backmapping endocytic bud induced by STxB B subunit**. As a second example we study the formation of an endocytic bud induced by the B-subunit of STxB. STxB mediates internalization of the whole protein (A and B subunits) by binding to its natural receptor globotriaosylceramide (Gb3) at the plasma membrane of target cells. STxB can enter the cell by building an endocytic bud. On giant unilamellar vesicles containing DOPC and low concentration of Gb3, STxB induces narrow tubular membrane invaginations[43]. Previously, we used DTS to investigate the formation of membrane tubular invaginations upon STxB binding[23,24]. Using the same procedure, we generated a system containing 75 STxB proteins and 480 vertices. Due to the reduction in number of the proteins, instead of tubular invaginations as in ref. [24], a vesicular bud is formed (Fig. 4a).

For backmapping, we first obtain an appropriate rescaling factor. Here, the rescaling factor is obtained by considering the smallest area of a vertex to be equal to the projected area of a STxB protein that gives $\mathcal{R} \approx 6.7$ nm (ref. [23]). Using this, we generated corresponding TS's for upper and lower monolayers (Step 1). The obtained TS's required four consecutive Pointillism operations ($m = 4$) to generate enough points for lipid and STxB placement (Step 2). To backmap a protein, we first mapped the all-atom structure of STxB bound to membrane in ref. [23] to the Martini model (Martini2.2 and Dry Martini use the same mapping). Then the protein, and membrane segment underneath, were selected to be transferred to the inclusion positions on each TS (Step 3) (Schematically shown in Fig. 1). After protein placement, DOPC lipids were placed in the remaining available area of both monolayers (Step 4). The final system contains 75 STxB proteins, 1125 Gb3 lipids and 110,791 DOPC lipids. After an energy minimization, and 70 ns equilibration in NPT ensemble, an NPT MD simulation of 200 ns was performed using the Dry Martini forcefield[39] (Fig. 4b). As our goal was to generate stable CG structures we did not perform any longer simulations (for energy evolution, see Supplementary Fig. 1B). Then we asked whether our algorithm can also generate a stable

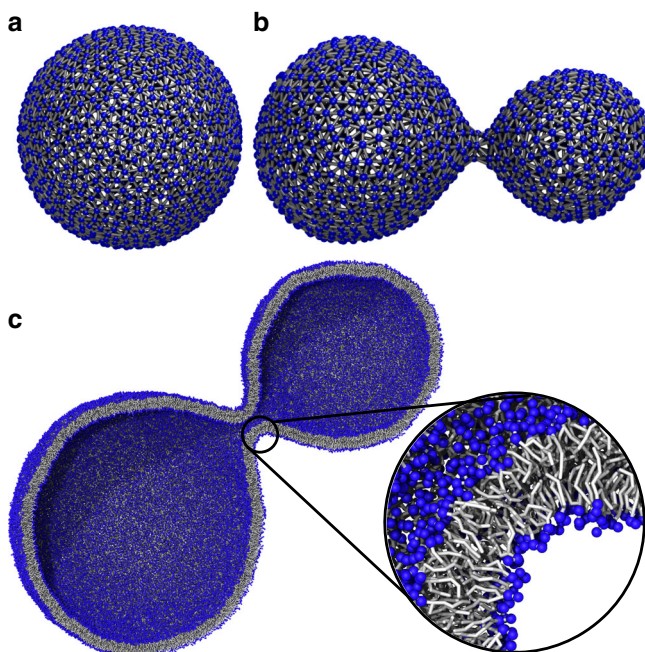

**Fig. 3 Backmapping of a growing vesicle. a** Original vesicle configuration in the DTS model. **b** Equilibrated vesicle structure after performing DTS simulation. **c** Backmapped vesicle to the Dry Martini model after 200 ns CG-MD simulation. Blue beads are phosphate group of the DOPC lipids, gray bonds represent the lipid tails.

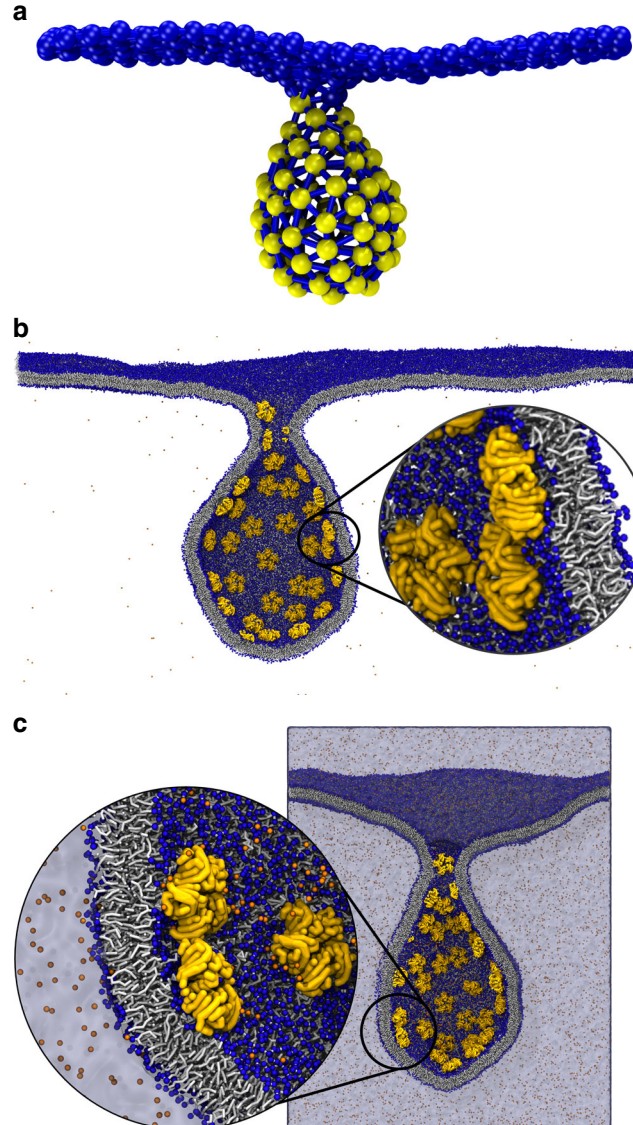

**Fig. 4 Backmapping of an endocytic bud formed by STxB proteins. a** Configuration obtained from DTS simulation. Yellow beads are vertices occupied by STxB inclusion, Blue beads are vertices representing the membrane surface. **b** Configuration from the solvent-free CG Martini model, after a 200 ns simulation starting from the backmapped DTS structure. Yellow objects are STxB proteins, orange dots are ions, blue beads are phosphate groups of the DOPC lipids and lipid chains are shown in gray. **c** Final CG configuration obtained with standard Martini model after a 50 ns simulation. The color code is the same as **b**.

structure for the standard Martini forcefield where there are explicit solvent particles. To do this, we started from the configuration obtained with the Dry Martini model, reducing the size of the system somewhat by removing part of the membrane far away from the membrane bud. The final box size was $97 \times 110 \times 110 \, nm^3$ containing 75 STxB proteins, 1125 Gb3 and 54,095 DOPC lipids. Subsequently, the system was solvated with more than $80 \times 10^6$ water beads and physiological salt concentration (Supplementary Table 1). We carried out energy minimization and equilibration of 40 ns, followed by a 50 ns MD simulation using the Martini2.2 forcefield. The final snapshot of the system is shown in (Fig. 4c), proving that our algorithm

produces a stable backmapped structure for the standard Martini model as well.

**Backmapping of an entire mitochondrion**. To further test the power of the method, we subsequently used TS2CG to build a Martini model of the membranes of an entire organelle with a realistic size and shape. For this, a triangulated surface of a mitochondrion was obtained from EM[44] (Fig. 5a). The dimensions of that system are $341 \times 488 \times 792 \, nm^3$. As we want to generate a mitochondrion with different lipid composition on each one of its bilayers (outer and inner membranes), we first disentangle the corresponding triangulated mashes of each of the two bilayers. Then a separate backmapping for each mash was performed with $\mathcal{R} = 1 \, nm$ as the TS's have the correct scaling. To generate points for lipid placement (Step 2) the outer membrane required five consecutive Pointillism operations ($m = 5$) while $m = 4$ was enough for the inner membrane as the TS of the inner membrane contains smaller size triangles. Figure 5b shows the resulting generated points for each of the monolayers. Step 4 was performed for both bilayers using a realistic mitochondrial lipid composition as obtained from experimental studies[45,46], and shown in Fig. 5c. For each lipid type in a specific monolayer, we obtained average APL values by performing symmetric bilayers simulations of the corresponding monolayer composition (Supplementary Table 2). Note that the trans-bilayer asymmetry in lipid composition can be realized directly using TS2CG. For more details about the composition of each of the monolayers, see Fig. 5c. Finally, CG structures of both bilayers were combined. The full system without any solvent molecules contains 83,288,300 particles, representing over six million lipids. After performing an energy minimization (as the system is large we first have performed energy minimization of several subsystems, for more details see Supplementary Fig. 2) and a short equilibration with position restraints on the lipid head groups, we carried out a short (2 ns) unrestrained molecular dynamics simulation of the entire mitochondrion using the Dry Martini forcefield. Within the 2 ns simulation, we did not observe a significant configurational change in the membrane shape (see Supplementary Fig. 3 for the total energy relaxation). However, for a longer run, the membrane is expected to deform since it does not include the necessary curvature inducing proteins to maintain this highly curved shape. The final configuration of the system is shown in Fig. 5c, demonstrating the power of TS2CG for generating large size and highly curved membranes.

## Discussion

MD simulations of biomolecular systems are reaching the length scale of their corresponding experimental systems[47]. This is mostly due to the rapid increase in available computational resources and efficient parallelization algorithms. However, the time-scale gap between MD simulations and experimental setups has remained a big challenge for two major reasons. Firstly, software parallelization becomes inefficient as the number of the allocated particles to each processing unit decreases. This means that longer simulations become strongly dependent on the computational power of a single core that is not increasing as rapidly. Secondly, the relaxation or equilibration time of many biological systems, where large-scale correlations exist, do not scale linearly with the system size (for instance lipid membranes shape fluctuations). Moreover, for large complex biological systems including cell membranes, the very different configurational structures are energetically degenerate and require very long simulations for proper sampling[2,17]. Mesoscale models such as DTS simulations are optimized to perform large-scale conformational changes of complex membranes (at the expense of

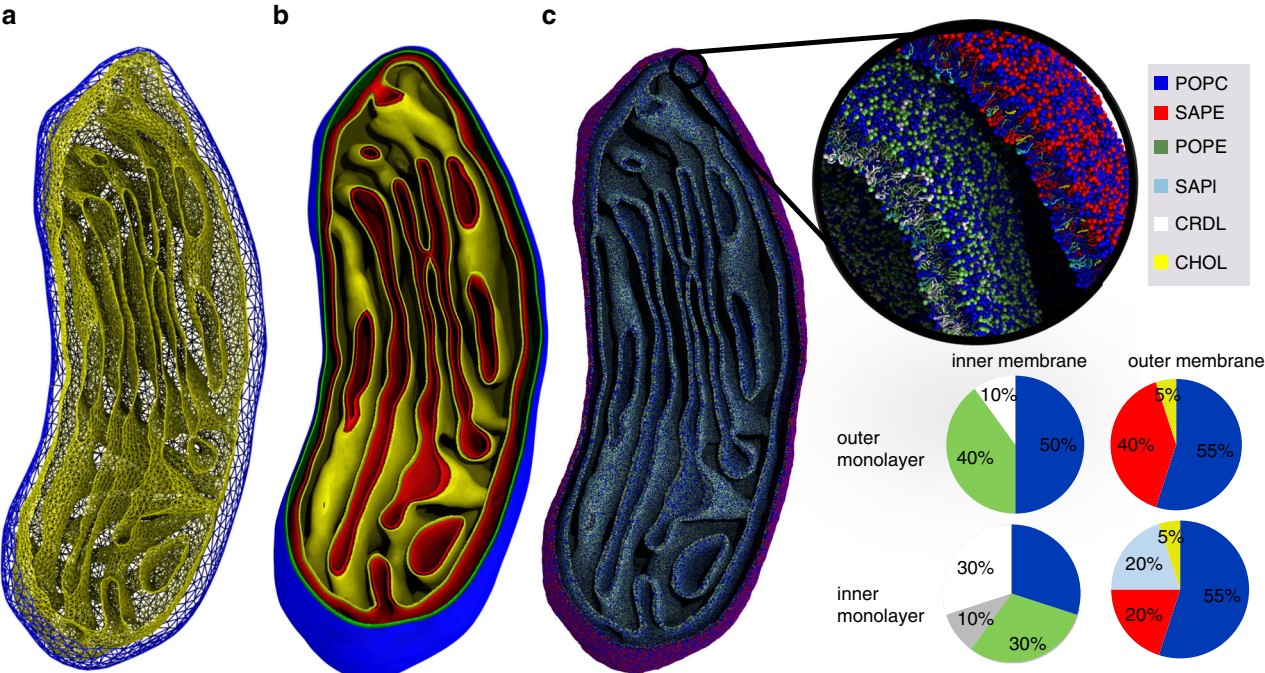

**Fig. 5 Transformation of mitochondrial membranes to near-atomic resolution. a** Cross section of the triangulated surfaces obtained from the EM map, separating outer (blue) and inner (yellow) membranes. **b** Cross section of the generated surface points for the outer monolayer of outer membrane (blue), inner monolayer of outer membrane (green), outer monolayer of inner membrane (red), and inner monolayer of inner membrane (yellow). **c** Backmapped structure after a 2 ns MD simulation with the Dry Martini force field. The zoomed views show the different lipid types, the overall composition is specified in the pi charts. POPC 1-palmitoyl-2-oleoyl-sn-glycero-3-phosphocholine, POPE 1-palmitoyl-2-oleoyl-sn-glycero-3-phosphoethanolamine, SAPE 1-stearoyl-2-arachidonoyl-sn-glycero-3-phosphoethanolamine, SAPI 1-stearoyl-2-arachidonoyl-sn-glycero-3-phosphoinositol, CHOL cholesterol, CRDL cardiolipin.

excluding many molecular details), and therefore a successful coupling between DTS and CG models could resolve the timescale challenge.

In this work, we described TS2CG, an algorithm that can backmap equilibrated structures of a mesoscale DTS simulation to CG molecular configurations while retaining the geometrical properties of the TS surface. As the membrane shape conformation is determined by its elastic energy that can be described as a function of the surface geometry, which is retained by TS2CG, no further large-scale equilibration will be required. We used TS2CG to backmap two structures obtained with DTS to the CG Martini model, and the results show that TS2CG generates configurations that are stable in subsequent simulations with both standard Martini and solvent-free Martini. In principle, TS2CG is not limited to backmapping to the Martini force field. TS2CG can also be exploited to generate large-scale complex membranes with different shapes (Supplementary Fig. 4) for any CG model by providing a proper TS structure and a CG model lipid library[34,35]. To build an all-atom structure, a two steps backmapping scheme can be used, in which a DTS output is first backmapped to a CG model and, after some equilibration, to an all-atom representation[48]. For example, TS2CG can be used for temporal enhancement of interactions between viruses, vesicles, and nanoparticles with a membrane[49,50] by using DTS up to engulfment or the prefusion state and then backmap to a CG or all-atom model for final equilibration.

To further increase the strength of coupling DTS to CG, one would require the DTS simulation to recalibrate itself in the course of the simulations based on information extracted from the CG level. In particular protein–protein interactions, as they can be different on curved membranes and furthermore depend on local lipid environment. To do this, an additional algorithm to map a CG configuration to a DTS model is required. A possible

scheme is to generate a TS using all the lipids head groups as vertices and then step by step merging every four neighboring triangles into a single one until the average triangle edge size (TS link length) reaches the rescaling factor ($\mathcal{R}$). An alternative route is to use DTS vertices as ghost particles in the CG simulations, that only interact with the membrane core[51] while their dynamics are mostly affected by the DTS potentials. There are a few elegant approaches that couple macroscale to microscale simulations of planar membranes[52,53], however, these do not consider the effect of membrane shape and the fluctuations on membrane-protein organization. The CG/DTS combination can resolve this limitation as both methods correctly capture these long-range effects.

TS2CG can also be used to couple macro-scale DTS simulations to high-resolution molecular simulations that focus on a specific sub-region of interest. The main difficulty in such cases is the introduction of periodic boundary conditions (PBC), which are typically necessary to avoid finite-size effects. In principle, there are two solutions to deal with this difficulty: the triangular mesh of the selected region may be adapted at the edges (or embedded) prior to backmapping to make the resulting system compliant with the PBC or the particles at the edges may be position-restrained (harmonically) to minimize the effects that may arise due to direct interactions of the solvent with the lipid tails. Of these, the first approach is suitable for maintaining compartmentalization. It is noted that with either approach, the membrane tension and local curvature may differ from the ones of the full system.

We also showed that TS2CG can be applied to TSs obtained in other ways, i.e. other than from a DTS simulation. In the example provided above, the TS was derived from an EM density map of an entire mitochondrion. Methods for converting this kind of data into a TS mesh are becoming more optimized[54]. With TS2CG, we could convert the outer and inner mitochondrial

membranes into full molecular resolution, with realistic lipid composition of each of the four lipid leaflets. Again, the back-mapped structure allowed subsequent simulation with the Martini force field, though the simulated time scale of such a huge system (more than 80 million particles) remains limited. With dedicated resources, however, and taking advantage of current developments in improving molecular dynamics software for very large systems, simulations in the microsecond time range are feasible that would allow to study the curvature-induced sorting of lipids inside the mitochondrial membranes. TS2CG furthermore allows insertion of mitochondrial membrane proteins such as the respiratory chain complexes for which Martini models are already available[55], opening the way to building—and simulating—a complete molecular model of this organelle. A related tool, called Lipidwrapper[35], can also generate molecular models for arbitrary shaped membrane surfaces, by stitching flat membrane patches together. However, this is problematic for a subsequent MD run as it requires large-scale membrane shape equilibrations, in particular for highly curved areas. TS2CG solves this issue by directing each lipid along the normal of an interpolated surface that inherits the curviness of the full system. In the current implementation, TS2CG only backmaps discretized surfaces, but it can be further extended for discretized material[56].

TS2CG can still be improved in a number of ways. In the current implementation of TS2CG, the APL of a specific molecule type is considered to be constant for all the regions of the curved membrane. Lipids are therefore distributed homogeneously across the surface. However, the equilibrium APL as well as the local lipid concentration can be dependent on the value and direction of the membrane curvature[27,51,57]. A solution to this limitation is to bias the acceptance probability ($P_{acc}$; see Step 4 of the Backmapping algorithm section) with the local membrane curvature, since TS2CG allows us to obtain the principal membrane curvatures at each point. Besides, the principal curvature direction can be used to place the lipids in the correct in-plane orientations. However, this requires a sufficient understanding of the relation between lipid packing and principal membrane curvature, which is currently sparse in the scientific literature[51]. The same is true for membrane proteins. Additionally, for membranes with asymmetric composition between leaflets, there will be additional challenges, as matching the inner and outer areas does not guarantee a tensionless state[58–60].

In conclusion, TS2CG opens a perspective to tackle the time-scale gap between MD simulations and experiments for membrane-bound systems. Our tool enables dual resolution CG/DTS simulations, with DTS accounting for the equilibration of large-scale degrees of freedom while CG models handle local moves of the constituent chemical components. In the near future, we expect TS2CG can be applied to whole cell simulations of realistic shape and composition.

## Methods

**Molecular dynamics simulation set up**. All the CG-MD simulations were performed using the GROMACS package (versions 2018 and 2019)[61,62]. Time steps of 30 fs for pure lipid systems and 20 fs for the protein system were used. Verlet cut-off scheme was used for both wet and Dry Martini simulations in which both van der Waals (cutoff) and Coulomb interactions (reaction-field) were set to zero at 1.1 nm[63]. Coulomb interactions were screened by a relative permittivity constant of 15.

Solvent-free simulations were performed using the Dry Martini force field[39] that uses the second-order stochastic dynamics (SD) integrator[64] in GROMACS with the friction constant of 4.0 ps. Dry Martini simulations were performed in NVT (for vesicle and mitochondrion systems) and NPT (for bud formation system) ensembles (Supplementary Table 1) and the temperature was kept constant at 310 K. For the NPT ensemble, the box was fixed in the $Z$ direction by setting the compressibility to 0 and the membrane was made tensionless by setting the reference pressure in the plane of the bilayer and compressibility to 0 and $3 \times 10^{-4}$ bar$^{-1}$, respectively. For equilibration, the Berendsen barostat[65,66] and for the final run the Parrinello–Rahman barostat[67] were used.

Simulations with explicit solvent were performed using the Martini 2.2 forcefield[40] and velocity-rescaling thermostat with time constant of the 1.0 ps (ref. [68]). The Berendsen barostat was used for equilibration with a reference pressure of 1 bar, a time constant of 4 ps and an isothermal compressibility of $3 \times 10^{-4}$ bar$^{-1}$. During production run, Parrinello–Rahman barostat with a time constant of 12 ps was used for pressure coupling.

The backmapped structures were first energy minimized by using the soft-core potential with initial $\lambda = 0.01$, $p = 2$ and $\alpha = 4$. Then, a normal (hard-core) energy minimization was performed. During the energy minimization, the DOPC phosphate beads and the protein backbone beads were restrained. For the Dry Martini system, after the energy minimization, an NVT simulation of 10 ns was performed, while DOPC phosphate beads and the proteins backbone beads were restrained. Then a 70 ns simulation, with the Berendsen barostat, was performed (no position restraint was applied). *Note*: For the vesicle system, only a 10 ns NVT simulation was performed. For the wet Martini system, only an NPT simulation with the Berendsen barostat was performed.

**STxB in Martini**. Both the wet and Dry Martini force field cannot produce the correct interaction between STxB and its bound Gb3 lipids, therefore, we apply intermolecular bonds to get the value of the induced local membrane curvature similar to the all atom MD simulations. To back map the protein, we first mapped the all-atom structure of STxB bound to membrane in ref. [23] to the Martini model (Martini2.2 and Dry Martini use the same mapping) and performed a short equilibration run while the phosphate group of the DOPC lipids and the protein backbone beads were restrained. Then the protein, and membrane segment underneath, were selected to be transferred to the inclusion's positions on the TS. STxB is a charged protein (5e$^-$), therefore, we neutralized the system by adding 375 Na$^+$ ions.

**Preparation of mitochondrion Martini model**. The triangulated surface for the mitochondrion was provided by Alexander Skupin, from the University of Luxembourg as a Blender mesh file, which was processed to yield a file readable by TS2CG software. Details on the procedure for triangulation from the raw EM data can be found in ref. [44]. We aimed at generating mitochondrion membranes with composition shown in Fig. 5c main text. To do this using TS2CG, we need to obtain APL for each lipid type at the given composition. For this, we performed MD simulations of four symmetric bilayers, that each contains the composition of each of the monolayers of the different mitochondrial membranes. Then the trajectories were analyzed using APL@voroni software[69] to obtain the APL for each lipid. The results are shown in Supplementary Table 2.

The mitochondrial membrane system is a very large system containing over 80 million particles. For a faster and easier minimization, we first divided the system into 20 subsystems (Supplementary Fig. 2). Then each subsystem was energy minimized using soft, and subsequently, hard-core potentials. Finally, the subsystems were recombined, and the full system was energy minimized again.

**Dynamically triangulated surfaces simulation**. In this method, the large-scale conformational properties of a fluid membrane are modeled by a dynamically triangulated, self-avoiding surface subject to PBC in the $X$ and $Y$ directions of the rectangular frame. The Euler characteristics of the surface are zero and therefore the number of vertices $N_v$, triangles $N_T$, and links between vertices $N_l$ are fixed and $6N_v = 3N_t = 2N_l$. The self-avoidance is ensured by enforcing the minimum and maximum tether length between neighboring vertices to be $d$ and $\sqrt{3}d$, respectively, in which $d$ is the length metric of the simulation. Using a set of discretized geometrical operations, a normal vector ($\hat{\mathbf{N}}_v$), an area ($A_v$) and two principal curvatures ($C_1(v)$, $C_2(v)$), and principal directions ($\mathbf{X}_1(v)$, $\mathbf{X}_2(v)$) are assigned to each vertex. The bending energy of the system is described by discretized form of the Helfrich Hamiltonian as

$$E_b = \frac{\kappa}{2}\sum_1^{N_v}(2H_v)^2 A_v - \kappa_G \sum_1^{N_v} K_v A_v \quad (3)$$

where $H_v = 0.5(C_1(v) + C_2(v))$ and $K_v = C_1(v)C_2(v)$ are mean and gaussian curvature, respectively, and $\kappa$ and $\kappa_G$ are bending rigidity and Gaussian rigidity. The membrane tension is fixed by a tension-controlling algorithm described in ref. [15]. A protein is modeled as a curvature inducing inclusion that can set on a vertex and modify its energy by

$$e_v = -\kappa H C_0 A_v \quad (4)$$

$C_0$ is the local curvature induced by an STxB protein. STxB–STxB interaction is modeled by energy term as

$$\varepsilon_{ij} = -2.5 + \left(1 + \cos 5\left[\Theta_i - \Theta_j'\right]\right) \quad (5)$$

if the proteins were located on two neighboring vertices and zero otherwise[23,24]. $\Theta_j'$ represents the orientation of inclusion residing of vertex $j$ after parallel transport to vertex $i$.

The equilibrium properties of the systems are analyzed by Monte Carlo simulation techniques with four MC moves, i.e., vertex move (a vertex is moved in a random direction), Alexander move (a mutual link between neighboring triangles

is flipped and two new triangles will be generated) and Kawazaki moves (an inclusion jumps to a neighboring vertex) and the membrane projected area change. To each Monte Carlo Sweep (MCS) with a probability of $P = 1/(5N_v)$, the membrane projected area is updated and with probability of $1 - P$, trial moves corresponding to $N_v$ vertex moves, $N_l$ Alexander move, and $N_i$ Kawazaki moves are performed (for more details, see refs. [15,24]). In all simulations, the bending rigidity was set to $\kappa = 20k_BT$ and $5 \times 10^6$ MCS were performed to reach final configurations.

**Reporting summary**. Further information on research design is available in the Nature Research Reporting Summary linked to this article.

## Data availability
Data supporting the findings of this manuscript are available from the corresponding authors upon reasonable request. A reporting summary for this Article is available as a Supplementary Information file.

## Code availability
The TS2CG source code, a user manual and tutorials are available on the Martini toolbox on github (https://www.github.com/marrink-lab/TS2CG).

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

## Acknowledgements

The authors thank John. H. Ipsen for constructive discussions and comments, Alexander Skupin from Luxembourg University for providing us with the triangulated mitochondrial surface and Bart Bruininks for assisting with rendering the very large system. This research is supported by the "BaSyC—Building a Synthetic Cell" Gravitation grant (024.003.019) of the Netherlands Ministry of Education, Culture, and Science (OCW) and the Netherlands Organization for Scientific Research (NWO).

## Author contributions

S.J.M., W.P., and T.A.W. conceived the study. W.P. and S.J.M. wrote the manuscript. M.K. and W.P. performed the simulations and analysis. W.P. created the software. All authors discussed the results and commented on the manuscript at all stages.

## Competing interests

The authors declare no competing interests.
