## [Peer Review File · Nature Communications]

Reviewers' comments:

Reviewer #1 (Remarks to the Author):

Overall this is an excellent manuscript on what I think is a game-changing tool; it will be highly useful in taking molecular level simulations to the next level in terms of system size. I have no problem with the methods - they are demonstrated in the manuscript to work.

My comments are minor, but ones I feel should be addressed. They concern the context of the present work. The authors provide comparisons to other approaches and indicate how theirs is better, but I feel the references to 50 and 51 are confusing - are you saying that the CG/DTS implementation is better than the methods employed in these papers? If so please explain how. Secondly it would be good to give examples of how current CG studies in the literature would benefit from using this approach, for example Jefferies and Khalid J Mol Biol 2020 already showed membrane undulations around vesicles, without CG-DTS, what would DTS add here? This is not a criticism of the present paper but an opportunity to show what extra the method could do. Also could the method be applied to the Fluctuating Finite Element approach of S.A. Harris for example see PloS Comp Biol, 2018.

Finally, it would be good to have a comment on how portions of the CG system could then be converted to all-atom detail for fine-grained studies of small patches.

Overall, I think this is an excellent & timely manuscript that will be undoubtedly be very useful and highly cited.

Reviewer #2 (Remarks to the Author):

Overall, this manuscript describes a timely and necessary tool to build CG models from triangulated meshes. The developers have created an algorithm that improves upon existing lipid packing software such as LipidWrapper. By using a pointillism approach and constructing the membrane with monolayers, rather than bilayers, the algorithm accounts for the increased number of lipids in the outer leaflet compared to the inner leaflet, and thus reduces the amount of time it would otherwise take to equilibrate a curved structure. The authors show (in at least one case) that their method is able to produce stable conformations ready for production runs. The work provides a much needed tool for computational biologists / biophysicists / chemists. The figures are clear, the data support the conclusions. The software was provided for review during the review process.

Considerations for additional elaboration in the manuscript:

- 1) Do the authors plan to maintain and continue to develop the code?
- 2) Could the authors please elaborate on the file types output by DTS and thus required as inputs for their algorithm?
- 3) Could the authors elaborate on ease of user application, whether the user must modify the scripts with the appropriate parameters (such as APL) or whether the parameters are executed as arguments in the command line?
- 4) The authors describe using an EM tomography map to create their triangulated mesh. Could the authors elaborate on the method of triangulation they used to generate their mesh from the tomography, and how easily the program could be modified to accept other triangulated mesh files (such as: dae, stl, or obj)?
- 5) Could the algorithm be easily modified to generate monolayer (rather than bilayer) structures?
- 6) The authors claim that they were able to achieve "stable" structures ready for simulation with the MARTINI model. For the mitochondrial membrane model, the authors do not comment on whether this resulting configuration was stable. Could the authors provide evidence that the

structure was able to achieve an energy-minimized conformation? Additionally, is the system also stable if the headgroup restraints are removed? It seems like a model this complex with such high variation in curvature would not be stable with short equilibrations and a very short production (2ns). If the system does not achieve a stable conformation, could the authors explain how the system evolves and any additional precautions users should take when working with systems like this? Could they also provide a general discussion of the challenges and additional considerations of working with bilayer systems of "unprecedented" size and shape?

7) There is a brief discussion on membrane heterogeneity and curvature. Did the authors then use an average APL to build the complex systems with heterogeneous membrane compositions? Did these membranes take longer to converge? Was there any evidence that the variation in APL caused local deformations from the starting mesh or the formation of pores?

Figure 1 does not have any lettering (A-E) as suggested by the text.

In the SI-F: first paragraph, final sentence: "backmapped" is misspelled as "backmaped"

This review was provided by Rommie E. Amaro and Abigail Dommer.

Reviewer #3 (Remarks to the Author):

In their manuscript "Backmapping triangulated surfaces to coarse-grained membrane models" Pezeshkian et al. describe, with three examples, a computational tool they call TS2CG. The tool takes as its input a triangulated surface in 3D and outputs a particle-based description of a biomembrane with a corresponding 3D geometry. The latter can then be used for molecular dynamics simulations of the biomembrane.

I strongly believe that software developers should be given the credit they deserve. However, I also think that the TS2CG tool does not represent an advance in scientific understanding that is likely to influence thinking in the field. To thus end, I do not find the manuscript suitable to be published in Nature Communications. More suitable venues would be, e.g., the journals SoftwareX or Molecular Simulation.

With regards to the TS2CG tool itself, I would like to draw the authors' attention to the following two points:

1) Currently the tool matches the areas of the two leaflets. This, however, does not guarantee that the preferred local curvature of the CG model agrees with that of the triangulated surface, because the area difference of the leaflets does not in fact predict the preferred curvature of the membrane, see Biophys. J. 115 1638 (2018), Nano Lett. 19 5011 (2019), and Biophys. J. 118 624 (2020).

2) Instead of a one-way script to map a triangulated surface to CG, it would be greatly more impactful to create a two-way mapping of the two Hamiltonians, such that the CG simulation would have a way of influencing the triangulated surface simulation and vice versa. Connecting the two simulations in such a rigorous way into a true multiscale simulation would be highly desirable.

We would like to thank the reviewers for their thorough reading of our manuscript, their suggestions and criticisms. We have carefully considered the points raised and modified the manuscript accordingly. We hope that our responses are fulfilling. The referee comments are in *italic* and our responses are in **bold. All the modified texts in the main manuscript and the Supplementary information are highlighted in red.**

Additionally, the following publications have been added to the list of references:

- 1) Jefferies, D. and S. Khalid, To infect or not to infect: molecular determinants of bacterial outer membrane vesicle internalization by host membranes. *J Mol Biol*, 2020. 432(4): p. 1251-1264.
- 2) Durrant, J.D., et al., Mesoscale All-Atom Influenza Virus Simulations Suggest New Substrate Binding Mechanism. *ACS Cent Sci*, 2020. 6(2): p. 189-196.
- 3) Solernou, A., et al., Fluctuating Finite Element Analysis (FFEA): A continuum mechanics software tool for mesoscale simulation of biomolecules. *PLoS Comput Biol*, 2018. 14(3): p. e1005897
- 4) Miettinen, M.S. and R. Lipowsky, Bilayer Membranes with Frequent Flip-Flops Have Tensionless Leaflets. *Nano Lett*, 2019. 19(8): p. 5011-5016.
- 5) Hossein, A. and M. Deserno, Spontaneous Curvature, Differential Stress, and Bending Modulus of Asymmetric Lipid Membranes. *Biophys J*, 2020. 118(3): p. 624-642.
- 6) Doktorova, M. and H. Weinstein, Accurate In Silico Modeling of Asymmetric Bilayers Based on Biophysical Principles. *Biophys J*, 2018. 115(9): p. 1638-1643.
- 7) Lee, C.T., et al., An Open-Source Mesh Generation Platform for Biophysical Modeling Using Realistic Cellular Geometries. *Biophys J*, 2020. 118(5): p. 1003-1008.

Reviewer #1 (Remarks to the Author):

Overall this is an excellent manuscript on what I think is a game-changing tool; it will be highly useful in taking molecular level simulations to the next level in terms of system size. I have no problem with the methods - they are demonstrated in the manuscript to work.

We would like to thank the referee for the positive opinion on our manuscript.

My comments are minor, but ones I feel should be addressed. They concern the context of the present work. The authors provide comparisons to other approaches and indicate how theirs is better, but I feel the references to 50 and 51 are confusing- are you saying that the CG/DTS implementation is better than the methods employed in these papers? If so please explain how.

Sorry for the confusion. Indeed, the implementations described in ref 50 and 51 are very elegant and powerful. We did not comment on the implementation nor did we compare our approach to them. We were merely highlighting the fact that the membrane shape is an important factor in the organization of the proteins and therefore needs to be considered in the mentioned approaches. We were emphasizing that for dual resolution approaches the CG-DTS combination can capture this missing feature.

We now modified the sentence and it reads as:

“There are a few elegant approaches that couple macro to micro scale simulations of planar membranes [52, 53], however, these do not consider the effect of membrane shape and the fluctuations on membrane-protein organization. The CG/DTS combination can resolve this limitation as both methods correctly capture these long-range effects.”

Secondly it would be good to give examples of how current CG studies in the literature would benefit

from using this approach, for example Jefferies and Khalid J Mol Biol 2020 already showed membrane undulations around vesicles, without CG-DTS, what would DTS add here?

We thank the referee for pointing this out. We have added a sentence and cited the mentioned paper that reads as:

“TS2CG can be used for temporal enhancement of interactions between viruses, vesicles and nanoparticles with a membrane [49, 50] by using DTS up to engulfment or the pre-fusion state and then backmap to a CG or all-atom model for final equilibration.”

This is not a criticism of the present paper but an opportunity to show what extra the method could do. Also could the method be applied to the Fluctuating Finite Element approach of S.A. Harris for example see PloS Comp Biol, 2018.

We thank the referee for pointing this out. We have added a sentence and cited the mentioned paper that reads as:

“In the current implementation, TS2CG only backmaps discretized surfaces, but it can be further extended for discretized material (space) such as[56]”

Finally, it would be good to have a comment on how portions of the CG system could then be converted to all-atom detail for fine-grained studies of small patches.

We thank the reviewer for pointing this out. We agree with the reviewer that this might be an important use case and we should have explicitly mentioned this.

“TS2CG can also be used to couple macro-scale DTS simulations to high-resolution molecular simulations that focus on a specific sub-region of interest. The main difficulty in such cases is the introduction of periodic boundary conditions (PBC), which are typically necessary to avoid finite-size effects. In principle, there are two solutions to deal with this difficulty: the triangular mesh of the selected region may be adapted at the edges (or embedded) prior to backmapping to make the resulting system compliant with the PBC or the particles at the edges may be position-restrained (harmonically) to minimize the effects that may arise due to direct interactions of the solvent with the lipid tails. Of these, the first approach is suitable for maintaining compartmentalization. It is noted that with either approach, the membrane tension and local curvature may differ from the ones of the full system.”

Overall, I think this is an excellent & timely manuscript that will be undoubtedly be very useful and highly cited.

Reviewer #2 (Remarks to the Author):

Overall, this manuscript describes a timely and necessary tool to build CG models from triangulated meshes. The developers have created an algorithm that improves upon existing lipid packing software such as LipidWrapper. By using a pointillism approach and constructing the membrane with

monolayers, rather than bilayers, the algorithm accounts for the increased number of lipids in the outer leaflet compared to the inner leaflet, and thus reduces the amount of time it would otherwise take to equilibrate a curved structure. The authors show (in at least one case) that their method is able to produce stable conformations ready for production runs. The work provides a much needed tool for computational biologists / biophysicists / chemists. The figures are clear, the data support the conclusions. The software was provided for review during the review process.

We thank the reviewers for their positive opinion of our work.

Considerations for additional elaboration in the manuscript:

1) Do the authors plan to maintain and continue to develop the code?

The code will be made available as part of the Martini tool-box on github (<https://www.github.com/marrink-lab>), where we and others can maintain and further develop the program. Planned development includes optimization of the user interface and implementation of adaptations required for selecting regions and making these compliant with PBC.

We know have clarified this in the Supplementary information:

**“The code will continue to be available as part of the Martini tool-box on github and will be further developed (<https://www.github.com/marrink-lab>). The planned development includes optimization of the user interface and implementation of adaptations required for selecting regions and making these compliant with PBC. Interested users are welcome to join us in the development.
“**

2) Could the authors please elaborate on the file types output by DTS and thus required as inputs for their algorithm?

TS2CG reads text files. We have added a discussion on the format of these files in the Supplementary information and a more extend version in the user manual.

3) Could the authors elaborate on ease of user application, whether the user must modify the scripts with the appropriate parameters (such as APL) or whether the parameters are executed as arguments in the command line?

It is user friendly to a high extent, TS2CG consists of two scripts, i.e., “Pointillism” (performs step 1 and 2, and is compiled into an executable binary file with the name PLM) and “CG membrane builder” (performs step 3 and 4 and is compiled into an executable binary file with the name PCG). PLM is command-line operated, while PCG uses a very simple input file and a few options from the command line. The program is generic and can be used as is for other CG models, with no need for changing the source code. All the options have been described thoroughly in the user manual which will be updated upon any new information.”

A short description has been added to the Supplementary information.

4) *The authors describe using an EM tomography map to create their triangulated mesh. Could the authors elaborate on the method of triangulation they used to generate their mesh from the tomography, and how easily the program could be modified to accept other triangulated mesh files (such as: dae, stl, or obj)?*

The triangulated surface for the mitochondrion was provided by Alexander Skupin, from the University of Luxembourg as a Blender mesh file, which was processed to yield a file in the format required for PLM. Details on the procedure for triangulation from the raw EM data can be found in the cited paper. The conversions from other file types of triangulated surfaces are typically straightforward, and we plan on providing an auxiliary (Python) script, rather than incorporating that conversion in the current program, which is written in C.

This is now clarified in the Supplementary information as:

“The triangulated surface for the mitochondrion was provided by Alexander Skupin, from the University of Luxembourg as a Blender mesh file, which was processed to yield a file in the *.q format (see above). Details on the procedure for triangulation from the raw EM data can be found in [12].”

5) *Could the algorithm be easily modified to generate monolayer (rather than bilayer) structures?*

Thanks for the suggestion. Yes indeed. We have added this feature in the latest version.

6) *The authors claim that they were able to achieve “stable” structures ready for simulation with the MARTINI model. For the mitochondrial membrane model, the authors do not comment on whether this resulting configuration was stable. Could the authors provide evidence that the structure was able to achieve an energy-minimized conformation? Additionally, is the system also stable if the headgroup restraints are removed? It seems like a model this complex with such high variation in curvature would not be stable with short equilibrations and a very short production (2ns). If the system does not achieve a stable conformation, could the authors explain how the system evolves and any additional precautions users should take when working with systems like this? Could they also provide a general discussion of the challenges and additional considerations of working with bilayer systems of “unprecedented” size and shape?*

Yes, the 2 ns simulation was performed without position restraints. We modified the sentence to clarify this and added the energy relaxation during the 2ns MD simulation as supporting figure in the Supplementary information.

“After performing an energy minimization (as the system is large we first have performed energy minimization of several subsystems, for more details see Supplementary information) and a short equilibration with position restraints on the lipid head groups, we carried out a short (2ns), unrestrained molecular dynamics simulation of the entire mitochondrion using the dry Martini forcefield”

Figure: Total energy relaxation of the mitochondrion model.

Up to the 2 ns simulation, we did not observe strong conformational changes in the membrane shape. However, this system is expected to change conformation as it does not include the curvature inducing proteins. The highly irregular mitochondrial membrane shape is the result of their binding. We have further clarified this point in the manuscript as

“Within the 2 ns simulation, we did not observe a significant configurational change in the membrane shape. However, for a longer run, the membrane is expected to deform since it does not include the necessary curvature inducing proteins to maintain this highly curved shape.”

7) There is a brief discussion on membrane heterogeneity and curvature. Did the authors then use an average APL to build the complex systems with heterogeneous membrane compositions?

Yes, for each lipid type an average APL was used in each monolayer. The mitochondrial membrane studied here is built homogeneous as we randomly placed each lipid type on the surface. We agree that for more realistic membranes this may not be the case and it can become heterogeneous, especially when the proteins are also considered. We have discussed this challenge in the discussion (see also referee 3 comment 1).

We now have clarified this point in the manuscript by adding the following sentence:

“For each lipid type in a specific monolayer, we obtained average APL values by performing symmetric bilayers simulations of the corresponding monolayer composition (see Supplementary information).”

Did these membranes take longer to converge? Was there any evidence that the variation in APL caused local deformations from the starting mesh or the formation of pores?

For the mitochondrial membrane, we do not believe nor claim that the lateral organization of the membrane has been converged. As the complexity of a membrane increases, the convergence time is expected to increase. However, at this point, we do not have any data to support this.

We have data on vesicular systems in which either a smaller or larger APL was initially used. Indeed, in these cases we observe some conformational changes within 100 ns simulations, leading to highly deformed regions including formation of pores.

To heal such defects, we typically apply first an equilibration procedure with position restraints on the lipid headgroups to avoid excessive deformations. After release of these restraints, pores typically seal spontaneously.

We have modified the manuscript to address this:

“After energy minimization, using the solvent free Dry Martini force field[39], and 10 ns of equilibration (using position restraints on the phosphate beads to avoid large size pore formation), a 200 ns MD simulation was performed (see SI for more detail on the MD simulations, including evolution of the system's energy Supplementary Figure 1A). The final structure of the MD simulation is shown in Figure 3-C. Note: within the first 50 ns, all the small pores were closed.”

Figure 1 does not have any lettering (A-E) as suggested by the text.

In the SI-F: first paragraph, final sentence: “backmapped” is misspelled as “backmaped”

Thanks for pointing this out. Both errors have been corrected in the new version.

This review was provided by Rommie E. Amaro and Abigail Dommer.

Reviewer #3 (Remarks to the Author):

In their manuscript "Backmapping triangulated surfaces to coarse-grained membrane models" Pezeshkian et al. describe, with three examples, a computational tool they call TS2CG. The tool takes as its input a triangulated surface in 3D and outputs a particle-based description of a biomembrane with a corresponding 3D geometry. The latter can then be used for molecular dynamics simulations of the biomembrane.

I strongly believe that software developers should be given the credit they deserve. However, I also think that the TS2CG tool does not represent an advance in scientific understanding that is likely to influence thinking in the field. To thus end, I do not find the manuscript suitable to be published in Nature Communications. More suitable venues would be, e.g., the journals SoftwareX or Molecular Simulation.

The referee seems to focus mostly on the program, while the core here is a new algorithm, based on computational geometry and membrane physics, allowing scientists to start exploring new directions in modeling complex systems and processes on multiple levels of resolution. With this approach to integrating mesoscopic DTS simulations with coarse grain (and atomistic) molecular simulations, we believe that we provide new views and allow new thoughts on the organization, dynamics and control of cellular membrane-based systems in a unified view from the protein to the organelle scale.

With regards to the TS2CG tool itself, I would like to draw the authors' attention to the following two points:

1) Currently the tool matches the areas of the two leaflets. This, however, does not guarantee that the preferred local curvature of the CG model agrees with that of the triangulated surface, because the area difference of the leaflets does not in fact predict the preferred curvature of the membrane, see *Biophys. J.* 115 1638 (2018), *Nano Lett.* 19 5011 (2019), and *Biophys. J.* 118 624 (2020).

TS2CG does not match the upper or lower monolayer area unless it is defined by the user. In an example described in the manuscript, i.e., the mitochondrion simulation, the inner and upper monolayer APL for the same lipid type is chosen differently. Nevertheless, we agree with the referee and are aware of the fact that matching the upper and inner areas does not guarantee a tensionless membrane. Additionally, APL for the same lipid type can be different in the same leaflet. We already had a section in the discussion on this issue (see below). However, exploring the relation between APL and membrane curvature and asymmetric bilayers is beyond the scope of the current manuscript. We do note that TS2CG can produce the correct structure if this information is provided.

“In the current implementation of TS2CG, the area per lipid of a specific molecule type is considered to be constant for all the regions of the curved membrane. Lipids are therefore distributed homogeneously across the surface. However, the equilibrium area per lipid as well as the local lipid concentration can be dependent on the value and direction of the membrane curvature [27, 49, 53]. A solution to this limitation is to bias the acceptance probability (P_{acc} ; see STEP 4 of Backmapping scheme section) with the local membrane curvature since TS2CG allows us to obtain the principal membrane curvatures at each point. Besides, the principal curvature direction can be used to place the lipids in the correct in-plane orientations. However, this requires a sufficient understanding of the relation between lipid packing and principal membrane curvature, which is currently sparse in the scientific literature [49]. The same is true for membrane proteins. “

We also added a new section to address the point raised by the referee more clearly:

“Additionally, for membranes with asymmetric composition between leaflets, there will be additional challenges, as matching the inner and outer area does not guarantee a tensionless state[58-60]”

2) Instead of a one-way script to map a triangulated surface to CG, it would be greatly more impactful to create a two-way mapping of the two Hamiltonians, such that the CG simulation would have a way of influencing the triangulated surface simulation and vice versa. Connecting the two simulations in such a rigorous way into a true multiscale simulation would be highly desirable.

Indeed, this will be the perspective for DTS, and we will consider such an approach in our next work. We had a large paragraph on this issue in the discussion section of the previous and current version of the manuscript that reads as:

“To further increase the strength of coupling DTS to CG, one would require the DTS simulation to recalibrate itself in the course of the simulations based on

information extracted from the CG level. In particular protein-protein interactions, as they can be different on curved membranes and furthermore depend on local lipid environment. To do this, an additional algorithm to map a CG configuration to a DTS model is required. A possible scheme is to generate a TS using all the lipids head groups as vertices and then step by step merging every four neighboring triangles into a single one until the average triangle edge size (TS link length) reaches the rescaling factor (R). An alternative route is to use DTS vertices as ghost particles in the CG simulations, that only interact with the membrane core [49] while their dynamics are mostly affected by the DTS potentials. An obvious benefit of CG/DTS combination over other coupled macro-micro scale modelling methods that are developed for planar membranes [50, 51], is that long-range effects of membrane shape and its undulations are correctly captured.”

REVIEWERS' COMMENTS:

Reviewer #1 (Remarks to the Author):

All of my points have been more than adequately addressed.

Reviewer #2 (Remarks to the Author):

The authors have done a great job replying to reviews and the paper is suitable for publication.